# UEChecker: Detecting Unchecked External Call Vulnerabilities in DApps via Graph Analysis

## Abstract

The increasing number of attacks on the contract layer of DApps has resulted in economic losses amounting to $66 billion. Vulnerabilities arise when contracts interact with external protocols without verifying the results of the calls, leading to exploit entry points such as flash loan attacks and reentrancy attacks. In this paper, we propose UEChecker, a deep learning-based tool that utilizes a call graph and a Graph Convolutional Network to detect unchecked external call vulnerabilities. We design the following components: An edge prediction module that reconstructs the feature representation of nodes and edges in the call graph; A node aggregation module that captures structural information from both the node itself and its neighbors, thereby enhancing feature representation between nodes and improving the model's understanding of the global graph structure; A Conformer Block module that integrates multi-head attention, convolutional modules, and feedforward neural networks to more effectively capture dependencies of different scales within the call graph, extending beyond immediate neighbors and enhancing the performance of vulnerability detection. Finally, we combine these modules with Graph Convolutional Network to detect unchecked external call vulnerabilities. By auditing the smart contracts of 608 DApps, our results show that our tool achieves an accuracy of 87.59% in detecting unchecked external call vulnerabilities. Furthermore, we compare our tool with GAT, LSTM, and GCN baselines, and in the comparison experiments, UEChecker consistently outperforms these models in terms of accuracy.

## 1 Introduction

The total value locked in decentralized applications (DApps) and encrypted protocols has exceeded $4.4 billion, attracting more than 82 million active cryptocurrency users worldwide(dappradar, 2024). By embedding established logic into smart contracts and storing them on the blockchain, these contracts can execute automatically without the need for centralized institutional intervention. This emerging technology allows users to autonomously control their personal funds, benefiting from the security and transparency provided by smart contracts. As blockchain and cryptocurrencies mature, DApps are rapidly expanding into various sectors, including finance (Angeris et al., 2021), gaming, NFTs, and social media. However, the threats posed by smart contracts are also becoming increasingly severe, such as flash loan attacks (Chen et al., 2024b) and market manipulation.

DApps demand extremely high security for smart contracts (Xia et al., 2021; Duan et al., 2022). Traditional methods such as symbolic execution (Luu et al., 2016; So et al., 2021), static analysis (Feist et al., 2019; Ghaleb et al., 2023), and taint tracking (Ghaleb et al., 2022) can quickly detect vulnerabilities but rely heavily on expert knowledge, often leading to false positives and consuming substantial computational resources (Wang et al., 2021; Durieux et al., 2020). In the context of DApps, where multiple contracts interoperate to achieve complex business logic, deep learning models have proven effective in identifying potential attack behaviors (Gao, 2020).

However, detecting vulnerabilities in DApps still presents the following challenges. Dynamic invocation: DApps are composed of multiple contracts involving numerous modules and branches. Different scenarios may arise when contracts are actually invoked, and this uncertainty in code logic

execution paths makes it difficult to ensure the security of external contract calls. Node function diversification: Different nodes represent different functional roles, such as external function calls, external call checks, and event triggers. This requires extracting features of nodes and call edges from a functional perspective. To address these issues, we propose UEChecker, a novel DApp security auditing tool. This tool utilizes Surya to extract high-level semantic representations of call graphs from source code and then identifies comprehensive graph information through node and edge feature extraction, clarifying the call relationships between different nodes.

The main contributions of this paper are as follows:

- To the best of our knowledge, we are the first to propose UEChecker, utilizing source code function body information to construct control and data flow call graphs to detect unchecked external call vulnerabilities in DApps. UEChecker first converts source code into call graphs leveraging Surya, and then analyzes and learns the features of the call graph to detect unchecked external call vulnerabilities in DApps.

- We validate the effectiveness of UEChecker in 608 DApps. Through auditing the vulnerability reports in DAppSCAN, we label contracts that explicitly involve unchecked external call vulnerabilities. We also conduct comparative analysis using baseline models and our proposed model, and experimental results demonstrate that our model achieves a higher detection accuracy of 87.59%, addressing the security issues of the complex vulnerabilities.

## 2 MOTIVATION AND PRELIMINARY

In this section, we primarily introduce the motivation behind our technology selection and the types of vulnerabilities that this paper focuses on Unchecked External Calls Vulnerabilities in DApp.

### 2.1 MODEL SELECTION

Compared to other advanced models for detecting contract vulnerabilities, GCN demonstrates unique advantages in identifying unchecked external call vulnerabilities in DApps (Li et al., 2024). GCN effectively processes information from various graph data structures, particularly for Dapps involving multi-contract calls. Smart contracts can be generalized into graph structures, where functions or variables can be viewed as nodes, and the edges represent the calling relationships between functions or variables. Through convolution operations, GCN captures both local and global information of nodes in the graph domain. For GCN, a single node can be represented as an aggregation of the entire graph's information, as shown in the formula $H^{(l+1)} = \sigma\left(\tilde{A}H^{(l)}W^{(l)}\right)$. This formula illustrates that each node in the graph is processed simultaneously, capturing more complex graph information features. Traditional models like RNNs and LSTMs require transforming graph-structured data into sequential data, necessitating the ordered aggregation of sequential information. This transformation often leads to the loss of graph data. The core of GCN lies in updating a node's global representation through neighborhood aggregation, a mechanism that balances information from the node and its neighboring nodes, thereby capturing the graph's contextual information in a higher dimension. It is crucial in handling unchecked external call vulnerabilities, as the occurrence of vulnerabilities relates to modifications of functions and variables across multiple contracts.

### 2.2 VULNERABILITY CHARACTERISTICS

Unchecked external call vulnerabilities typically occur when a smart contract neglects to check the return value after calling an external function, allowing malicious contracts to exploit the vulnerability. Dapps consist of multiple contracts with complex calling chains, which can be intuitively represented as graphs. The connections between nodes are embedded into the feature representations. GCN can receive features from nodes and their neighbors through each layer's convolution operation, ensuring that the contextual information of the vulnerability is effectively captured by GCN. Hierarchical information aggregation helps GCN extract features of unchecked external call vulnerabilities. For such vulnerabilities, GCN gradually aggregates and extracts all nodes' features. By learning a weight matrix, GCN performs weighted learning on different neighboring nodes, abstracting the vulnerability features from the graph. The graph feature of the $K^{th}$ layer contains both

global and local information of all nodes from the k-1 layers. The logic of the vulnerability is crucial in identifying unchecked external call vulnerabilities. GCN can detect this vulnerability because it captures global graph information at a higher level through its multi-layer structure, enabling a comprehensive analysis of the logic where the vulnerability occurs. This capability is lacking in other traditional tools like symbolic execution, taint analysis, and formal verification.

### 2.3 DApp Unchecked External Calls Vulnerabilities

Previous works (Liu et al., 2022; Brent et al., 2020; Chen et al., 2024a; Zhong et al., 2024) clearly explain the unchecked external calls vulnerabilities that have occurred in smart contracts. Most types of DApps involve token transfer operations. In the development of DApps, attack points often emerge in functions related to token transfers and external calls, such as transfer(), send(), call(), and delegatecall(). When designing these functions, it is crucial to carefully check the return values of the calls. The Unchecked External Calls vulnerability is a serious flaw caused by smart contracts calling external functions without verifying the return values.

The DApp contract in Listing 1 is a simplified version derived from the real world. This contract has an Unchecked Calls Return Value vulnerability, identified as a high-risk issue by SlowMist during the audit of the Booster Protocol. The vulnerability occurs in the safeTokenTransfer() function(line 22), where the transfer return value is not checked. As a result, it is impossible to determine if the transfer was successful. When the _token.transfer() call(line 26) fails without throwing an exception, the contract assumes that the funds were successfully transferred and proceeds with subsequent operations. For instance, in the Reward() function(line 15), when a user calls Reward() to claim their reward, if the transfer within safeTokenTransfer() fails but the external call is not checked, the reward becomes invalid. However, the contract incorrectly assumes the transfer was successful.

---

Listing 1: Real-world contract with unchecker external call vulnerability

```
1   contract RewardPool is Ownable{
2       using SafeMath for uint256;
3       IERC20 public rewardToken;
4       mapping(address => uint256) public rewards;
5       constructor(address _rewardToken) public{
6           rewardToken = IERC20(_rewardToken);
7           }
8       function depositRewards(uint256 amount) external onlyOwner{
9           rewardToken.transferFrom(msg.sender, address(this), amount);
10          }
11      function Reward() external{
12          uint256 reward = rewards[msg.sender];
13          require(reward > 0, "No rewards to claim");
14          rewards[msg.sender] = 0;
15          safeTokenTransfer(msg.sender, reward);
16          }
17      function safeTokenTransfer(address _to, uint256 _amount) internal{
18          uint256 balance = rewardToken.balanceOf(address(this));
19          uint256 value = _amount > balance ? balance : _amount;
20          if (value > 0) {
21              rewardToken.transfer(_to, value); // Unchecked Call Return
                    Value
22          }
23      }
24  }
```

---

We validated Listing 1 using three advanced smart contract detection tools, and the results indicate that Oyente, Confuzzius, and Conkas cannot detect this vulnerability in practice. Symbolic execution excels at detecting specific paths, but the unchecked external call vulnerability is a logical flaw, with the vulnerability point not being on the path. Oyente does not delve into return value checks, making it unable to identify this vulnerability. Similarly, Confuzzius and Conkas cannot logically analyze the occurrence point of the unchecked external call vulnerability because Confuzzius relies on specific inputs to trigger the vulnerability and does not effectively check return values.

Listing 2: Uniswap Flash Loan Attacks Reported in Previous Works

```
1  contract UniswapV2Pair {
2      function swap(uint amount0Out, uint amount1Out, address to, bytes
           calldata data) external {
3          require(amount0Out > 0 || amount1Out > 0, 'UniswapV2:
               INSUFFICIENT_OUTPUT_AMOUNT');
4          (uint balance0, uint balance1) = getReserves();
5          uint amountIn = amount0Out > 0 ? balance0 : balance1;
6          require(amountIn > 0, 'UniswapV2: INSUFFICIENT_INPUT_AMOUNT');
7      }
8  }
```

Listing 2 shows a flash loan attack on the Uniswap platform, which fails to adequately check the results of an external call. The attacker borrowed a large amount of money and executed trading operations to take advantage of price fluctuations for arbitrage. The point of attack occurred when the external call return value was not checked in the getReserves() function(line 4).

## 3 METHODOLOGY

The architecture of the UEChecker, as illustrated in Figure 1, is composed of two primary modules.

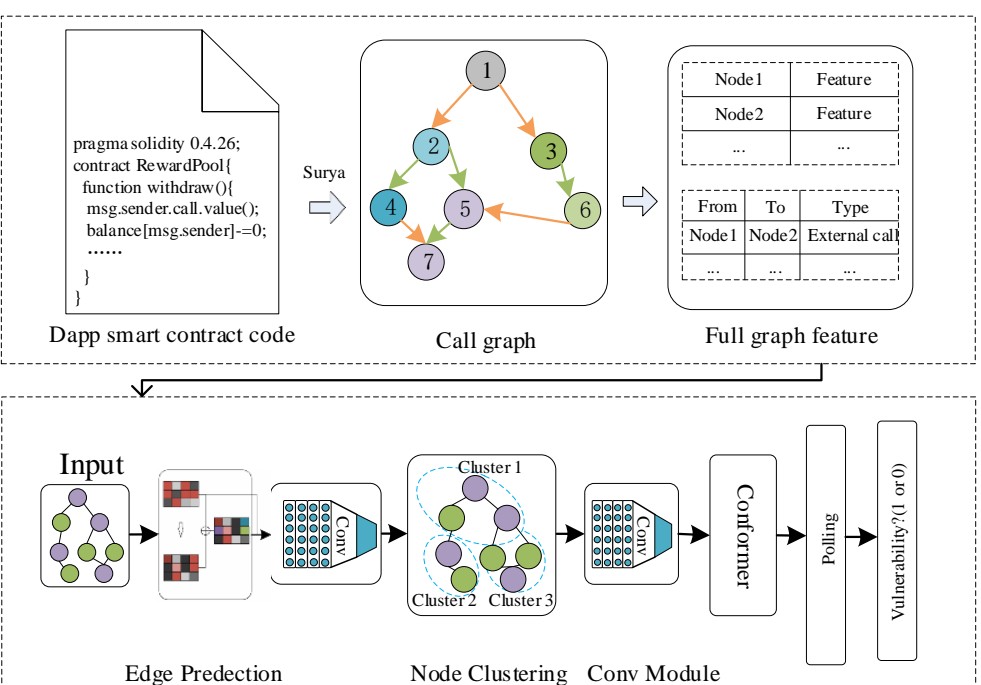

Figure 1: Overview of the Framework. The tool is divided into two phases. The first phase is the call graph feature information extraction, and the second phase is the composition of the model layers.

In the first part, UEChecker transforms the source code of smart contracts into a high-performance call graph semantic representation using Surya and extracts the features of nodes and edges in the smart contract by parsing the call graph. The second part learns the features of the graph to identify unchecked external call vulnerabilities. The Algorithm that describes the process from feature extraction to vulnerability detection is shown in the Appendix A.2.

### 3.1 CODE TO CALL GRAPH FEATURE

#### 3.1.1 CODE TO DOT

The tool utilizes the practical utility Surya to transform the source code of dApp contracts into a call graph structure. First, Surya performs lexical analysis on the source code, breaking it down into lex-

ical units (operators, keywords, etc.), followed by syntactic analysis to construct an abstract syntax tree (AST) while identifying all function declarations and marking the call relationships between functions. Finally, by constructing function dependencies, each function is represented as a node, with calls between nodes generalized as edges, where each edge represents the dependency between functions, thereby determining the call graph. By analyzing the call graph, we can understand the interactions and calls between functions within the contract and between different contracts. This provides deeper insights into the code execution flow and the message transmission between nodes, facilitating the extraction of nodes and the information between calling edges. The call graph allows for a rapid understanding of the contract's functionality and structure (Feist et al., 2019).

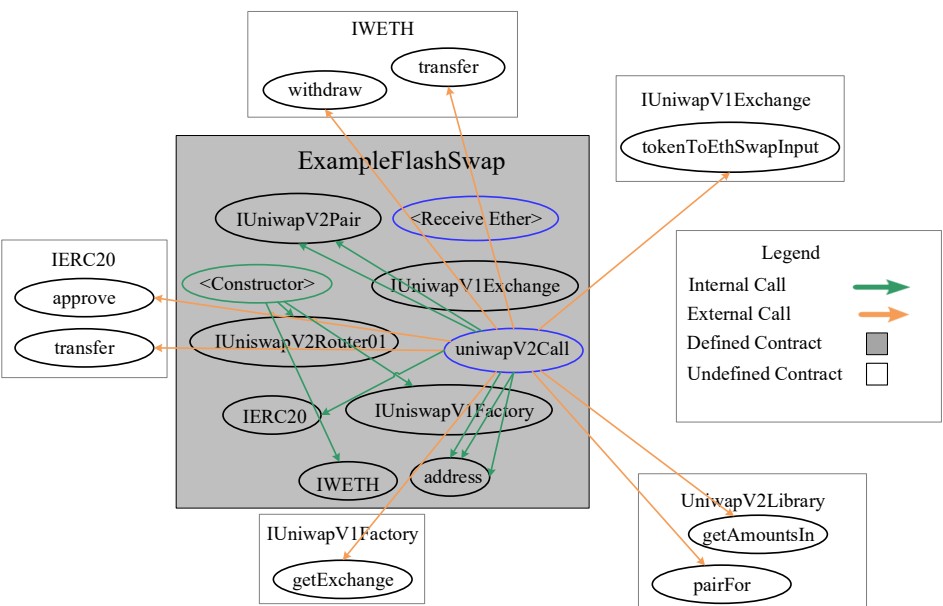

Figure 2: Each rectangle represents a contract. If a function within a contract calls a function in another contract, an arrow of a different color is used to indicate this interaction.

The call graph model generated using Surya is shown in Figure 2. It involves sensitive operations such as transfers, approvals, and withdrawals, which are external contract calls. External call edges are distinguished by different colors. When a contract function calls an external function, the return value can be checked through the label on the edge between nodes. The feature information of contract nodes is obtained by embedding node names and labels into a vector space. All the dot format call graphs are generated using Surya. The graph feature matrix and adjacency matrix are created by reading all the dot files. The dot file contains the ID and label of each node, and the edge type is obtained by extracting the edge marker field information. An embedding matrix is created based on the label information, converting labels into indices and obtaining the corresponding graph node feature vectors based on the embedding matrix. The adjacency matrix and adjacency dictionary are constructed using the edge information, where the adjacency matrix represents the connectivity between nodes, and the adjacency dictionary is used for subsequent graph data normalization.

### 3.1.2 GRAPH FEATURE EXTRACTION

## 3.2 DETECTION MODELS

### 3.2.1 EDGE PREDECTION

Equation 1 describes the edge prediction process. The feature information for each pair of nodes is obtained based on graph features, and the adjacency matrix for each pair of nodes is reconstructed. A node pair is defined as $E(x_i, x_j)$, and the edge prediction network uses a linear layer and a batch normalization layer to predict edges for the set of nodes. The edge prediction neural network, denoted as $f_{\text{edge}}$, is formulated as follows:The edge prediction network then integrates $y_{ij}$ into the

adjacency matrix $A$, and the final adjacency matrix is obtained as $A = A + A^T$.

$$y_{ij} = \exp\left(0.5 \cdot \left(f_{\text{edge}}(x_i, x_j) + f_{\text{edge}}(x_j, x_i)\right)\right) \tag{1}$$

### 3.2.2 CLUSTERING COMPUTATION

Equations 2 and 3 show how to find clusters of clustered nodes by clustering centers. After edge processing, the nodes are passed through a graph convolutional network and then enter the node clustering layer. Nodes are repositioned based on the input node features $x$ and the cluster centers $C_k$. The distance between the sample node and the cluster center is calculated by Equation 3.

$$d_{ijk} = \sum_{c=1}^{C} (x_{ijc} - c_{kc})^2 \tag{2}$$

The clustering assignment of the sample is determined by the distance $d_{ijk}$, as shown below.

$$a_{ij} = \arg\min_k d_{ijk} \tag{3}$$

where $a_{ij}$ is the index of the cluster center to which the sample belongs.

### 3.2.3 CONFORMER BLOCK

Residual connections and layer normalization are added before and after each layer in the Conformer Block to capture the spatiotemporal information of the call graph sequence data. The Conformer Block integrates multi-head attention, convolutional operations, a feedforward neural network and other layers. 1) Feedforward Network implements a feedforward neural network using two linear layers, employing the GELU activation function and dropout mechanism to enhance the nonlinear expression capability of the module. 2) The multi-head self-attention mechanism focuses on the relationships between different parts of the graph features. The to_qkv linear layer maps the input data to query (Q), key (K), and value (V) vectors. The Equation 4 for the attention is as follows. The attention weights are computed using the scaled dot-product equations and normalized through the softmax function. The attention scores are then used to perform a weighted sum of the input features. 3) The scale layer scales the output of each sublayer using a scaling factor, thereby controlling the gradient propagation and the model's convergence speed. 4) PreNorm layer applies layer normalization before each sublayer to ensure that the input of each layer has a stable distribution.

$$\text{Attention}(\mathbf{Q}, \mathbf{K}, \mathbf{V}) = \text{softmax}\left(\frac{\mathbf{Q}\mathbf{K}^\top}{\sqrt{d_k}}\right)\mathbf{V} \tag{4}$$

### 3.2.4 CONVOLUTIONAL LAYER

We introduce the convolution operation to capture local spatiotemporal information in the graph, as Equation 5 shows. The Laplacian matrix is used to enhance the connectivity of the graph by expanding the adjacency matrix as $\hat{A} = I + A$, where $I$ is the identity matrix and $A$ is the adjacency matrix. Feature update is performed according to the number of adjacency matrices, $X_r = L_r \cdot x$, where $r$ denotes the number of adjacency matrices and $L_r$ is the Laplacian matrix computed from the adjacency matrix $A_r$ for each relation type, i.e., $\hat{A}$. Finally, all features are concatenated and represented through a fully connected layer. The convolutional layer is calculated as in Equation 6, where $H^{(l+1)}$ is the node feature matrix of the $l$ layer, $\tilde{A}$ is the normalized adjacency matrix, $W$ is the weight of the $l_{th}$ layer, and $\sigma$ means the ReLU activation function.

$$x_{\text{out}} = \left[(x \cdot (I + \text{adj\_sq} \cdot A_r))\right]_{r=0}^{R-1} \cdot mask \tag{5}$$

$$H^{(l+1)} = \sigma\left(\tilde{A}H^{(l)}W^{(l)}\right) \tag{6}$$

## 4 EXPERIMENTS

### 4.1 EXPERIMENTAL SETTINGS

All experiments are executed on a server equipped with NVIDIA GeForce GTX 4070Ti GPU, Intel(R) Core(TM) i9-13900KF CPU, and 128G RAM, operating on Ubuntu 22.04 LTS. The software environment includes Python 3.9 and PyTorch 2.1.1

### 4.1.1 DATASET

In our experiment, we use the most comprehensive DApps dataset currently available, DAppSCAN, to detect unchecked external call vulnerabilities in DApp smart contracts and to validate the effectiveness of our tool. DAppSCAN includes 608 projects across various categories such as DeFi and games, encompassing 21,414 smart contracts. Additionally, DAppSCAN provides audit reports for these DApps. We manually checked smart contracts involving unchecked external call vulnerabilities based on these audit reports. Since DApps incorporate a substantial number of external library references, the actual number of DApp smart contracts handling core business operations is 3,833 vulnerability-free contracts. Furthermore, we validate the smart contract dataset provided by Liu et al. (Liu et al., 2023), which contains 1,199 contracts marked as unchecked external call defects.

### 4.1.2 IMPLEMENTATION DETAILS

Our model is trained for 600 epochs using the AdamW optimizer. The batch size is set to 30, and the initial learning rate is 25x10e5. The attention layer uses 8 heads, each with a dimension of 64. The convolutional layer is equipped with ReLU activation and a linear layer with in_features X out_features. For the GCN, the hyperparameters include a hidden layer of 256, an edge hidden layer of 32, and a dropout of 0.2. Through a series of experiments, we found that our model is relatively insensitive to most hyperparameters, except for learning rate and hidden layer size. We experimented with a range of values, including learning rates 0.00015, 0.0002, 0.00025, 0.0003 and hidden layers 32, 64, 128, 256, 512, and found that a learning rate of 0.00025 and a hidden layer of 256 improved the model's accuracy after 600 epochs.

## 4.2 EVALUATION

We conduct experiments to address the following research questions: **RQ1:** Can UEChecker accurately identify unchecked external call vulnerabilities in the dataset? **RQ2:** Does UEChecker outperform other models in detecting unchecked external call vulnerabilities in DApps? **RQ3:** Do the edge prediction, node aggregation, and Conformer Block modules improve the detection performance of the model?

### 4.2.1 ANSWER TO RQ1

The loss function experimental results, as shown in Table 1, demonstrate the detection efficiency and performance of UEChecker. In DApps, external contract calls are first imported through the import directive, and then external functions are called. Our tool achieves an accuracy of 87.59% and a recall of 86.53%, effectively identifying external function calls and determining whether unchecked external calls are present. Upon examining DApp contracts with vulnerabilities, we find that mature interfaces, such as the ERC20 protocol in the OpenZeppelin library, have robust handling mechanisms. However, using such protocols still requires checking external calls. The precision of 90.30% reflects the model's accuracy in predicting positive samples, and the F1 Score, calculated from precision and recall, is 88.30%, providing a measure of the model's overall performance. Table 2 presents the performance under different loss functions. When comparing the performance of UEChecker with different loss functions (BCEWithLogitsLoss and CrossEntropyLoss), it is found that CrossEntropyLoss performs better in terms of F1-score (87.12%) and Precision (90.30%), indicating that it is more effective at reducing false positives and overall provides higher prediction accuracy. While BCEWithLogitsLoss achieves a higher Recall (90.30%), which demonstrates its ability to better identify positive samples, its lower Precision (63.75%) leads to more false positives. Therefore, CrossEntropyLoss is more suitable for the task of identifying unverified external call vulnerabilities in this context.

Table 1: Performance evaluation of UEChecker on Datasets

| Tool | Acc | Recall | Precision | F1-score |
|------|------|--------|-----------|----------|
| UEChecker | 87.59% | 86.53% | 90.30% | 87.12% |

Table 2: The Performance Comparison of Different Loss Functions.

| Network Structures | Loss Function | F1-score | Precision | Recall |
|---|---|---|---|---|
| UEChecker | BCEWithLogitsLoss | 72.84% | 63.75% | 90.30% |
| UEChecker | Cross Entropy | **87.12%** | **90.30%** | 86.53% |

### 4.2.2 ANSWER TO RQ2

Since DApps are not composed of single contracts and the logic processing involves multiple external functions, analyzing contracts using source code or bytecode alone is insufficient to compile or extract the bytecode of DApps. Thus, we developed comparative baseline models with other AI models and conducted comparison experiments with our tool. We reference models such as SaferSC by Tann et al. (2018), which is the first deep learning-based smart contract vulnerability detection model using an LSTM network to construct an Ethereum opcode sequence model, achieving precise smart contract vulnerability detection. We also reference DeeSCVHunter by Yu et al. (2021), which uses FastText embedding to convert code into vectors and builds a Bi-GRU model. Table 3 presents the detection results of various tools for unchecked external call vulnerabilities. When detecting unchecked external call vulnerabilities in DApps, the LSTM model aggregates only the contextual information of the call graph and overlooks function calls, resulting in an accuracy of only 60.3%.

Table 3: Performance Comparison of baseline models.

| Model | Acc | Recall | Precision | F1-score |
|---|---|---|---|---|
| LSTM | 60.30% | 77.48% | 59.97% | 72.97% |
| GAT | 48.82% | 42.59% | 43.35% | 36.60% |
| GCN | 52.93% | 72.74% | 49.86% | 57.51% |
| UEChecker | **87.59%** | **86.53%** | **90.30%** | **87.12%** |

Due to the increased complexity of the call graph structure in DApps, the generalized nodes and edges significantly increase, and the GAT model, being more sensitive to noise within the call graph, struggles with capturing long-distance function calls despite using attention mechanisms. This results in an accuracy of only 48.82%. The GCN model, composed of just two convolutional layers, also fails to accurately identify features spanning multiple nodes and paths, thereby weakening its ability to distinguish different node features, leading to an accuracy of only 52.93%. By incorporating edge aggregation modules, node aggregation modules, and conformer block modules, our model outperforms others across four key metrics: accuracy, recall, precision, and F1 Score, achieving an accuracy of 87.12% and also improving recall. The performance of UEChecker surpasses that of other models. From the comparative experimental results, we conclude that our proposed model excels in detecting unchecked external call vulnerabilities within DApp contracts, with detection accuracies improving by 27.29%, 38.77%, and 34.66%, respectively, compared to other models.

### 4.2.3 ANSWER TO RQ3

Table 4 summarizes the performance metrics obtained under different components. Comparison of model complexity and performance: GCN, as the most basic model, performs the weakest in terms of these metrics. Although Recall (72.74%) is relatively high, it only indicates that the model is able to identify more positive samples, while its low Precision (49.86%) suggests that many of the samples predicted as positive are actually negative, leading to a significant number of false positives. The overall F1-score (57.51%) is also the lowest among these model structures. The model with the addition of the Edge Prediction component shows a significant improvement in detection performance, with Recall increasing to 79.71%. The model performs better in detecting positive samples, while Precision also improves to 79.38%, resulting in a reduction in false positives. The F1-score increases significantly to 82.15%, indicating a good balance in overall performance. The addition of the Cluster module further improves Precision to 80.75%, enhancing the model's accuracy in detecting positive samples.

Table 4: The Performance Comparison of Different Composite Structures.

| Network Structures | Acc | Recall | Precision | F1-score |
|---|---|---|---|---|
| GCN | 52.93% | 72.74% | 49.86% | 57.51% |
| Edge Pred + GCN | 78.8% | 79.71% | 79.38% | 82.15% |
| Edge Pred + Cluster + GCN | 78.04% | 76.85% | 80.75% | 76.08% |
| UEChecker | **87.59%** | **86.53%** | **90.30%** | **87.12%** |

Table 5: Performance Comparison of Related Tools.

| Tool | Acc | Recall | Precision | F1-score |
|---|---|---|---|---|
| Oyente | 52.53% | 50.15% | 90.27% | 64.58% |
| Mythril | 34.89% | 47.34% | 50.26% | 48.75% |
| Security | 74.10% | 53.90% | 86.74% | 68.72% |
| Confuzzius | 53.43% | 53.44% | 66.03% | 59.07% |
| UEChecker | **87.59%** | **86.53%** | **90.30%** | **87.12%** |

The proposed model, which incorporates a Conformer Block on top of the previous structure, shows significant improvements in all metrics upon validation on the dataset. As shown in Table 5, accuracy reaches 87.59%, and Recall and Precision improve to 86.53% and 90.30%, respectively. With both high Precision and Recall, the F1-score also increases to 87.12%, making this the best-performing structure among all models.

## 5 DISCUSSION

Current advanced smart contract detection tools mainly rely on techniques such as static analysis, symbolic execution, and deep learning to identify contract vulnerabilities. These tools parse the source code, construct syntax structures, and execution paths to identify potential security flaws. However, as DApps increase in complexity due to rising demand, existing tools face challenges such as insufficient analysis depth, high false-positive rates, and difficulties in handling cross-contract call relationships. The call graph reveals the call dependencies between DApp contract functions, making it suitable for identifying complex call dependency relationships. Therefore, we extract call graph features to detect potential vulnerabilities within the call chain. In designing our vulnerability detection algorithm, we propose a method for detecting vulnerabilities in DApp smart contracts by capturing neighborhood node information in the call graph. We propose a GCN-based model for detecting vulnerabilities in DApp smart contracts. With an accuracy of 87.59% in detecting unchecked external call vulnerabilities in 608 DApps, our model effectively identifies these vulnerabilities.

## 6 CONCLUSION

We present a GCN-based deep learning framework for detecting unchecked external call vulnerabilities in DApps. We design three modules to capture graph neighborhood information, enhance feature representation between nodes, and improve the accuracy of detecting unchecked external call vulnerabilities in DApps. We use Surya to convert the source code into a call graph representation and detect vulnerabilities through comprehensive feature extraction. In comparative experiments, our tool achieves an accuracy of 87.59% on the dataset, surpassing the accuracy of our baseline model. It effectively identifies unchecked external call vulnerabilities in real-world decentralized applications.

## 7 REPRODUCIBILITY

To ensure the reproducibility of our study, we provide all necessary resources, including all datasets, full details of the experimental setup as outlined in Section 4, all code required for conducting experiments.

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

## A APPENDIX

### A.1 THE USE OF LARGE LANGUAGE MODELS (LLMS)

During the paper writing process, LLMs have been utilized for English translation and polishing.

### A.2 THE DETECTION ALGORITHM

The Algorithm 1 describes the process from feature extraction to vulnerability detection.

---

Algorithm 1: Identifying DApp vulnerabilities from source code

---

**Require:** source_files

  initialize a Graph and Feature G, F;

  **for** each $file \in source\_files$ **do**

    (nodes,edges) = surya(file);

    G.append(edges,ndoes);

  **end for**

  **for** each $g \in G$ **do**

    (gLable,(Ns,Ne,type),FCnames) = GraphInfo(g);

    Feature = convert2adj(gLable,(Ns,Ne,type),FCnames);

    F.append(Feature);

  **end for**

  B, N, C = loaddata(F)

  **for** each $b \in B$ **do**

    node_pair = fine_node_pair(b);

    x_cat = concatenate_feature(node_pari);

    y = edge_pred(x_cat);

  **end for**

  Y = construct_adj_matrix(y);

  data = combine(Y, F[1]);

  x = gcn(data)[0];

  x = dropout(x);

  x = cluster_layer(x);

  x = gcn(x);

  x = x * mask.unsqueeze(-1);

  x = conformer_block(x);

  x = dropout(x);

  x = max_polling(x);

  x = fullconnect(x);

  **return** x

---

