# OpenReview forum: "UEChecker: Detecting Unchecked External Call Vulnerabilities in DApps via Graph Analysis"
_ICLR.cc/2026/Conference — Submitted to ICLR 2026_

### Official Review · Reviewer_RshB · 2025-10-22

**Soundness:** 2
**Presentation:** 2
**Contribution:** 2
**Rating:** 2
**Confidence:** 4

**Summary:**

This paper proposes UEChecker, a graph-based deep learning framework to detect Unchecked External Call vulnerabilities in decentralized applications. The model extracts function-call graphs from Solidity source code using Surya, encodes node and edge features, and then applies a hybrid architecture combining Edge Prediction, Node Clustering, GCN, and a Conformer Block. The idea is to reconstruct missing graph connections, group semantically similar nodes, and capture multi-scale dependencies for better vulnerability detection.
Experiments on the DAppSCAN dataset show the proposed model outperforms traditional static-analysis tools and basic deep learning baselines.

**Strengths:**

- The paper addresses a real and security-critical problem in DApp ecosystems, namely unchecked external calls, which are hard to detect through symbolic execution.
- The method pipeline is easy to follow.
- Experimental results demonstrate solid improvements over traditional static tools and basic deep learning  models on DAppSCAN.

**Weaknesses:**

- The proposed approach combines well-known modules without introducing algorithmic novelty or theoretical insights.
- The baselines include only early neural network models and static tools, mitting all modern pretrained or LLM-based detectors.
- The reported performance superiority is not meaningful under current standards, where LLM-based methods routinely exceed 95–98% F1 on UEC or similar SWC vulnerabilities.
- Experiments are conducted only on DAppSCAN. There is no testing on other datasets or unseen contracts.
- The performance drop of Edge Pred + Cluster + GCN relative to Edge Pred + GCN is not analyzed. Likely, the clustering layer’s hard assignment may introduce information loss, but the paper does not provide any discussion.
- The paper contains numerous grammatical errors (e.g., leading to exploit entry points), typos (e.g., polling), and nonstandard mathematical expressions (e.g., $A = A + A^{T}$)

**Questions:**

- It is recommended that the authors include comparisons with modern pretrained and LLM-based models.
- The paper could be strengthened by explaining whether the proposed modules (GCN, Edge Prediction, Clustering modules) replaced or complemented by transformer-based reasoning mechanisms.
- It would be valuable for the authors to discuss the causes of the performance drop observed in the Edge Pred + Cluster + GCN setting.
- The contribution could be enhanced by incorporating qualitative case studies to illustrate how the model distinguishes vulnerability patterns.
- The paper would benefit from providing an evaluation on unseen or cross-dataset scenarios.
- Please improve the writings and figures.

---

### Official Review · Reviewer_3v41 · 2025-10-26

**Soundness:** 2
**Presentation:** 2
**Contribution:** 2
**Rating:** 2
**Confidence:** 3

**Summary:**

This paper introduces UEChecker, a deep learning-based tool for detecting unchecked external call vulnerabilities in decentralized applications (DApps). The proposed approach first constructs a call graph from smart contract source code to model the contracts' structure and interactions. It then employs a Graph Convolutional Network (GCN) enhanced with three key components—an edge prediction module, a node clustering module, and a conformer block module—to learn rich feature representations and capture complex dependencies within the graph. Evaluated on a dataset of 608 real-world DApps, UEChecker achieves an accuracy of 87.59%, demonstrating superior performance over multiple baseline models and established vulnerability detection tools.

**Strengths:**

1. The paper focuses on a critical and common vulnerability type in smart contract security. These vulnerabilities are frequently exploited to launch attacks like Reentrancy and Flash Loan attacks, bearing significant practical relevance and economic impact.

2. The paper presents a novel and well-structured approach by combining call graph analysis with a GCN architecture to target the specific problem of unchecked external calls. The integration of edge prediction, node clustering, and the conformer block, demonstrates a thoughtful design that effectively captures both local node relationships and global graph context, which is critical for accurately identifying this vulnerability pattern in complex DApps.

3. The authors provide extensive experiments, including performance comparison with multiple baseline models (LSTM, GAT, GCN) and traditional tools (Oyente, Mythril), as well as thorough ablation studies that convincingly validate the contribution of each proposed module.

**Weaknesses:**

1.The paper's primary contribution is one of application and engineering (building a tool for a specific security task) rather than methodological innovation in machine learning. The UEChecker model itself is a "kitchen-sink" combination of existing components (GCN, clustering, Conformer). The paper fails to provide a strong theoretical or empirical justification for this specific, and rather unusual, combination.

2. The description of the model's core components is severely underspecified. Most critically, the use of a Conformer—an architecture designed for sequential data—after GCN graph feature extraction is methodologically baffling and left entirely unexplained. This suggests a potential fundamental misunderstanding of the inductive biases of these architectures.

3. The reported performance of baseline models (e.g., GCN at 52.93% accuracy ) is abnormally low, suggesting they were not properly implemented or tuned. This calls into question the validity of UEChecker's claimed relative improvements.

4. The ablation study (Table 4 ) shows that adding the "Cluster" module degrades performance (F1-score drops from 82.15% to 76.08% ), yet this component is retained in the final model, a direct contradiction.

5.  The paper heavily relies on Surya for the fundamental step of call graph generation, yet provides no description, reference, or link to this tool.

6. The description of the model's core components is severely underspecified. Most critically, the use of a Conformer—an architecture designed for sequential data—after GCN graph feature extraction is methodologically baffling and left entirely unexplained. This suggests a potential fundamental misunderstanding of the inductive biases of these architectures.

**Questions:**

see weekness

---

### Official Review · Reviewer_J9uc · 2025-10-31

**Soundness:** 1
**Presentation:** 2
**Contribution:** 2
**Rating:** 4
**Confidence:** 2

**Summary:**

The authors present UEChecker, a deep learning framework for detecting unchecked external call vulnerabilities in DApp smart contracts. By combining call graph analysis, edge prediction, node aggregation, and a Conformer Block within a Graph Convolutional Network, UEChecker effectively captures complex contract dependencies. Experiments on 608 real-world DApps show that UEChecker achieves 87.59% accuracy, outperforming traditional baseline models.

**Strengths:**

1. **Significance of the Topic**

The paper addresses an important problem in the field of blockchain security—detecting unchecked external call vulnerabilities in DApp smart contracts.

**Weaknesses:**

1. **Limited Novelty:**

While the paper proposes a framework based on graph neural networks and convolutional modules, the approach of encoding program structure using graph-based convolution has been extensively explored in prior work (see, for example, Flow2vec [1]). The main novelty appears to lie in the edge prediction module, but the manuscript does not compare this component against relevant baselines, making it difficult to assess its effectiveness and actual contribution.

2. **Baseline Selection:**

The experimental comparison is limited to traditional models such as LSTM, GAT, and GCN. The paper does not include comparisons with more recent or domain-specific approaches that use similar graph-based or program analysis techniques, which limits the persuasiveness of the evaluation.

3. **Clustering Module Effectiveness:**

The effect of the clustering module is unclear. As shown in the ablation study, the performance actually decreases when the clustering module is added, raising questions about its necessity and justification in the overall framework.

**References:**

[1] Sui, Y., Cheng, X., Zhang, G., & Wang, H. (2020). Flow2vec: Value-flow-based precise code embedding. Proceedings of the ACM on Programming Languages, 4(OOPSLA), 1-27.

**Questions:**

See weaknesses, thanks.

---

### Official Review · Reviewer_QAw8 · 2025-11-01

**Soundness:** 2
**Presentation:** 3
**Contribution:** 2
**Rating:** 4
**Confidence:** 3

**Summary:**

This paper proposes UEChecker, a deep learning–based framework for detecting unchecked external call vulnerabilities in decentralized applications (DApps). The method transforms smart contract source code into call graphs using Surya, and then analyzes the structural and semantic dependencies between functions through multiple modules: an edge prediction module, a node aggregation module, and a Conformer Block that integrates multi-head attention and convolutional layers. The resulting graph representation is processed with a Graph Convolutional Network (GCN) to identify vulnerability patterns. Experimental results on 608 DApps show that UEChecker achieves high accuracy, outperforming baselines such as GCN, GAT, and LSTM in detecting this class of vulnerabilities.

**Strengths:**

1. The unchecked external call vulnerability is a critical issue in DApp security, and the paper provides strong motivation through real-world examples (e.g., Uniswap flash loan attacks).
2. The design of UEChecker’s modular architecture (edge prediction, clustering, Conformer integration) is clearly articulated and logically justified. The integration of GCN and Conformer blocks is an interesting approach to modeling multi-scale dependencies in call graphs.
3. The paper includes comparative experiments against multiple baseline models and demonstrates measurable improvements.

**Weaknesses:**

1. The framework primarily extends standard GCN architectures with additional processing modules; the conceptual innovation is incremental.
2. Evaluation is restricted to 608 DApps, which may not be sufficient to demonstrate generalization to diverse contract ecosystems.
3. The contribution of each module (edge prediction, clustering, Conformer) is not quantitatively isolated or validated.
4. While accuracy is reported, there is little analysis of specific error cases or robustness across contract structures.

**Questions:**

1. How sensitive is UEChecker’s performance to the structure of the call graphs (e.g., number of nodes, contract modularity)?
2. Have you evaluated how the model performs on unseen contract templates or obfuscated code?
3. Can you provide ablation results to show how each module (edge prediction, clustering, Conformer) contributes to final accuracy?
4. How does UEChecker handle large-scale DApps with hundreds of interconnected contracts, does inference time or memory usage become a bottleneck?
5. Is the model robust against adversarially modified call graphs (e.g., dummy nodes or reordered functions)?

---

### Meta-Review · Area_Chair_3uy8 · 2026-01-08

**Summary:**

The reviewers concerned about the incremental contribution, and limited evaluation settings.

**Reviewer Concerns:**

The reviewers concerned about the incremental contribution, and limited evaluation settings.

**Reviewer Scores:**

No reviewer will increase the score since no rebuttal is provided.

---

### Decision · Program_Chairs · 2026-01-26

Reject